# A Long-Term Energy-Rich Diet Increases Prefrontal BDNF in Sprague-Dawley Rats

**DOI:** 10.3390/nu14010126

**Published:** 2021-12-28

**Authors:** Alessandro Virtuoso, Pernille Tveden-Nyborg, Anne Marie Voigt Schou-Pedersen, Jens Lykkesfeldt, Heidi Kaastrup Müller, Betina Elfving, Dorte Bratbo Sørensen

**Affiliations:** 1Department of Veterinary and Animal Sciences, Faculty of Health and Medical Sciences, University of Copenhagen, DK-1870 Frederiksberg, Denmark; a.virtuoso@hotmail.com (A.V.); ptn@sund.ku.dk (P.T.-N.); amvoigt@gmail.com (A.M.V.S.-P.); jopl@sund.ku.dk (J.L.); 2Translational Neuropsychiatry Unit, Department of Clinical Medicine, Aarhus University, DK-8000 Aarhus, Denmark; heidi.muller@clin.au.dk (H.K.M.); betina.elfving@clin.au.dk (B.E.)

**Keywords:** BDNF, Cafeteria Diet, diet-induced obesity, experimental animal models

## Abstract

Findings of the effect of high-fat feeding including “Cafeteria Diets” (CAF) on brain-derived neurotrophic factor (BDNF) in the hippocampus (HIP) and prefrontal cortex (PFC) in rodents are conflicting. CAF is a non-standardized, highly palatable energy-rich diet composed by everyday food items for human consumption and is known to induce metabolic syndrome and obesity in rats. However, the highly palatable nature of CAF may counteract a negative effect of chronic stress on anticipatory behavior and synaptic plasticity in the hippocampus, hence represent a confounding factor (e.g., when evaluating functional effects on the brain). This study investigated the effects of a chronic, restricted access to CAF on BDNF, monoamine neurotransmitters, and redox imbalance in HIP and PFC in male rats. Our results show that CAF induced BDNF and its receptor TrkB in PFC compared to the controls (*p* < 0.0005). No differences in monoamine neurotransmitters were detected in either PFC or HIP. CAF increased dehydroascorbic acid and decreased malondialdehyde in PFC (*p* < 0.05), suggesting an early redox imbalance insufficient to induce lipid peroxidation. This study supports that a chronic CAF on a restricted schedule increases BDNF levels in the PFC of rats, highlighting that this may be a suboptimal feeding regime when investigating the effects of diet-induced obesity in the brain and emphasizing this as a point of attention when comparing the findings.

## 1. Introduction

The effect of diets rich in fat and sugar have been extensively studied in rodent models regarding their effects on physiology [1,2,3] and behavior, both in cognitive [4,5] and non-cognitive models [6,7]. Studies in rats have also highlighted how the consumption of a diet rich in saturated fats and sugar has a negative effect on both hippocampal-dependent cognition and levels of brain-derived neurotrophic factor (BDNF) in the hippocampus (HIP) [8,9,10,11]. BDNF is a member of the protein family neurotrophins, which has been shown to promote neuronal development in the cortex and HIP [12]. It is also involved in learning and memory, especially in HIP [13], and has been shown to be reduced during high-fat feeding regimens in rats and mice with subsequent reduced cognitive performance [9,10,14]. All these effects are mediated by the BDNF receptor tropomyosin receptor kinase B (TrkB) [15], whose fully glycosylated form is considered to be the mature and active form of the receptor [16,17]. Binding of BDNF to TrkB triggers different signaling cascades, some of which regulate protein translation and transport (mitogen-activated protein kinase MAPK, and phosphatidylinositol 3-kinase PI3K), whereas others regulate the intracellular release of calcium in postsynaptic sites (phospholipase Cγ, PLCγ) [18]. These downstream signaling pathways are initiated by the binding of BDNF to TrkB and autophosporylation of tyrosine residues including Y817, which serves as docking sites for effector proteins in the intracellular domain [19,20]. Thus, assessing the effects on BDNF, TrkB, and Y817 provides information on the potentially involved regulatory pathways as well as the putative biological significance (i.e., active vs. inactive forms).

Furthermore, it has been reported that high fat diets alter the dopamine metabolism in the brains of rats [21], and inflict a downregulation of the dopamine D2-receptor. This effect has been shown to be inversely correlated to body weight (i.e., high body weight reducing the expression of the receptor and consequently having a negative effect on dopamine signal transduction in the brain) [22]. Moreover, obesity has been associated with higher serotonin (5-HT) turnover, with elevated levels of its metabolite 5-hydroxyindoleacetic acid (5-HIAA) as well as of the dopamine metabolite homovanillic acid (HVA) in the cerebrospinal fluid of people with obesity [23].

The Cafeteria Diet (CAF) has been gaining attention as a robust model of the metabolic syndrome in rats [24]. Rodent studies have linked the consumption of CAF to a decrease in hippocampal BDNF and hippocampal inflammation in rats [4,25], and to increased lipid peroxidation in the hippocampus (HIP) in mice [26]. However, CAF is a non-standardized composite diet formed by several everyday items for human consumption from cookies, to sausages, crackers, butter, etc. Its composition varies widely between studies, which is mirrored in the behavioral results [27,28]. Furthermore, it has been suggested that restricted access to palatable food not only creates anticipatory behavior in rats [29], but also counteracts the negative effects of chronic stress on synaptic plasticity in rat HIP [30,31]. Moreover, in a previous study from our group, we showed that rats fed CAF adopted a more optimal behavioral strategy faster than controls in an operant conditioning paradigm [32]. Building on the two above-mentioned premises, we hypothesized that a long-term restricted access to CAF will result in the following: (1) An increase in the levels of BDNF and its receptor TrkB in the HIP and PFC of male rats; (2) an increase in the metabolism of monoamine neurotransmitters in the HIP and PFC of male rats; and (3) impose redox stress in the HIP and PFC of male rats.

The aim of this study was to highlight the complex and highly variable effect of a non-standardized diet on the brain of rodents, and its potential effect on behavioral and brain research as a confounding factor in chronic models of diet-induced obesity.

## 2. Materials and Methods

Forty male Sprague-Dawley (NTac:SD) rats (Taconic Biosciences Inc., DK-4623 Lille Skensved, Denmark) were seven weeks old at the time of arrival. Animals were housed three per cage in polypropylene cages (1354G Eurostandard type IV, 595 × 380 × 200 mm, Scanbur, Karlslunde, Denmark) with environmental enrichment items (cardboard tunnel, cardboard house, wood chewing block, nesting material) on a 12 h light cycle (lights on at 06:00). One cage in each group housed only two animals. As part of the experimental protocol, the animals were subjected to behavioral testing in the decision making paradigm during the study (once a month for five months, excluding training in the procedure; see “Behavioral procedures”). These data have been previously published [32]. The animals were assigned to either one of two groups using learning speed in the behavioral procedure as stratum. The groups received the following diets (Table 1): animals in the control group (n = 20) were fed regular rodent chow (Altromin 1324, Altromin, Lage, Germany) and tap water ad libitum, while those in the treatment (n = 20) group were fed the same diet as the controls plus a selection of cookies (Pally Biscuits BV, Nieuwegein, The Netherlands), cakes (Bisca A/S, Stege, Denmark), and butter (Arla Foods, Viby, Denmark), henceforth named the Cafeteria Diet (CAF), administered once a day for five days a week over the course of five months (Appendix A).

For CAF, the values refer to the averaged values of food items, as these were equally represented. Water content was calculated by placing samples of each diet at 50 °C for 8 h and weighing the remainders. CAF = Cafeteria Diet.

At euthanasia, animals were sedated with 0.3 mL/100 g body weight of a mixture of fentanyl citrate (0.315 mg/mL)/fluanisone (10 mg/mL) (Vetapharma Ltd., Sherburn in Elmet, Leeds, UK) and midazolam (B. Braun, Melsungen, Germany) (1:1) administered subcutaneously. Animals were then decapitated with a rat decapitator.

### 2.1. Behavioral Procedures

All animals of both groups were tested in an operant conditioning test, the decision making paradigm (DM). The test and its results are extensively reported elsewhere [32]. Briefly, the DM required the animals to make a decision between two alternative behavioral strategies (pressing a lever close to a water dispenser, and moving to break a photo-beam far from the dispenser) in order to obtain water rewards. A water-deprivation schedule was used to build motivation to work. The optimal solution of the problem was considered to be, following the marginal value theorem [33], the highest amount of rewards per lowest amount of work. Water-deprivation was applied to both groups in the same way, therefore, the potential stress deriving from it was considered to be uniform between groups. Similarly, the potential effects imposed by the cognitive challenge presented by the DM were considered to be uniform between the groups.

### 2.2. Tissue Sampling

At euthanasia, after decapitation, skulls were opened with an Aesculap bone cutter starting from the foramen magnum at an angle of approximately 30° from the sagittal crest until the lateral limit of the bregma. The parietal bone was then lifted, and the brain exposed, and gently extracted with surgical forceps. The brain was divided along the midline, HIP was identified in each hemisphere, gently removed with a pair of small-sized surgical forceps, and placed in a 1.5 mL Eppendorf tube and immediately stored on dry ice. Subsequently, a sample of the prefrontal cortex (PFC) was obtained by transversally cutting rostral to the third ventricle (olfactory bulbs were discarded). Samples were placed in a 1.5 mL Eppendorf tube and stored on dry ice until it was transferred to a −80 °C freezer.

### 2.3. BDNF and TrkB Receptor

All analyses were performed on the left hemisphere. HIP and PFC samples were homogenized using 4-mm stainless steel beads on a tissue mixer-mill (Retsch GmbH, Haan, Germany). The samples were mixed twice for 30 s at 30 Hz in 10× volume of cell lysis buffer (Bio-Plex™ Cell Lysis Kit, Bio-rad, Hercules, CA, USA) containing factor 1, factor 2, complete protease inhibitor cocktail (Roche, Basel, Switzerland), sodium fluoride (5 mM), and sodium metavadanate (1 mM). The samples were then allowed to freeze at −80 °C overnight before analysis. On the day of analysis, samples were sonicated (40% power, 4 s), and after centrifugation (4500× *g*, 4 min, 4 °C), the supernatants were transferred to new tubes. Total protein concentrations were determined using the Pierce^®^ BCA Protein Assay Kit (ThermoFisher Scientific, Roskilde, Denmark).

### 2.4. Western Blotting

HIP and PFC samples (20 µg total protein) were separated on 10% criterion TGX gels (Bio-Rad) and transferred to nitrocellulose membranes. The samples were then blocked in odyssey blocking buffer (LI-COR, Lincoln, NE, USA) and probed with the primary antibodies: rabbit anti-BDNF (Sigma AV41970; 1:2000), mouse anti-β-actin (926-42212, LI-COR; 1:3000), goat anti-TrkB (AF1494, R&D Systems, Minneapolis, MN, USA; 1:500), and rabbit anti-TrkB(Y817) (ab81288, Abcam, Cambridge, UK; 1:1000) overnight at 4 °C. This was followed by incubation with the appropriate IRDye conjugated secondary antibody for 1 h at RT: IRDye 800CW donkey anti-goat IgG, IRDye 680RD donkey anti-rabbit IgG, IRDye 800CW goat anti-rabbit IgG, or IRDye 680RD goat anti-mouse IgG, all in 1:15,000 dilution (LI-COR). Infrared signals were detected using the Odyssey CLx infrared imaging system (LI-COR, and bands were quantified using Image Studio software (LI-COR).

### 2.5. Oxidative Stress Markers and Monoamine Metabolites

All analyses were performed on the right hemisphere. Brain samples (HIP and PFC) were taken directly from −80 °C and homogenized in 4 °C Dulbecco’s PBS (pH 7.4). For analysis of ascorbic acid (AA) and dehydroascorbic acid (DHA), homogenates were stabilized with equal amounts of 10% (*w*/*v*) MPA in 2 mM Na2-EDTA and analyzed using high performance liquid chromatography (HPLC) with coulometric detection as described previously [34,35,36]. Measurement of lipidoxidation by malondialdehyde (MDA) and dihydrobiopterine (BH_2_) and tetrahydrobiopterine (BH_4_) was performed on HPLC as previously described [37,38]. Measurement of monoamine neurotransmitters dopamine, serotonin (5-HT), and norepinephrine as well as associated metabolites 5,4-dihydroxyphenylacetic acid (DOPAC), homovanillic acid (HVA), and 5-hydroxyindoleacetic acid (5-HIAA) was performed by HPLC as previously described [39]. Unless otherwise stated, samples were run in a blinded randomized fashion, in duplicates or more, together with standardized controls.

### 2.6. Statistical Analysis

All data were analyzed with GraphPad Prism 6 (GraphPad Software Inc., San Diego, CA, USA). The D’Agostino–Pearson omnibus test was employed to verify the normality of the distribution of all datasets. All datasets were analyzed by the unpaired t-test, except for body weight, which was analyzed with repeated measures 2-way ANOVA, with time and diet as factors. Holm–Šidak post hoc test was used to perform multiple comparisons between time points. For the western blot analyses of BDNF and TrkB levels in the left PFC and HIP, a subset of 12 random samples was selected from each group. Data for these analyses are presented as a percentage of control. All data are presented as mean ± standard deviation (SD).

## 3. Results

### 3.1. Body Weight

We found a significant effect of time (*p* < 0.0001) and diet (*p* = 0.004) on the body weight of animals, but also a significant interaction time * diet (*p* < 0.0001); this reflects previous studies employing similar diets [24]. Holm–Šidak’s multiple comparison test showed a statistically significant difference between groups starting at month two (*p* = 0.0307). The data have been previously published [32].

### 3.2. BDNF and TrkB Receptor

Results are presented in Figure 1. We observed significantly higher levels of BDNF in PFC of rats eating the CAF (% of control 129 ± 16.39; *p* = 0.0003; Figure 1A). Similarly, the total level of TrkB was higher in animals eating CAF compared to controls (% of control 116.7 ± 13.06; *p* = 0.0008; Figure 1B), and the difference carried on while examining its mature (i.e., fully glycosylated, % of control 119.0 ± 14.63; *p* = 0.001; Figure 1C) and immature (i.e., partially glycosylated, % of control 115.1 ± 14.67; *p* = 0.0059; Figure 1D) forms. Conversely, the ratio of phosphorylated tyrosine residue Y817 (pTrkB) to mature TrkB receptor was significantly lower in animals eating the CAF (% of control 86.55 ± 13.08; *p* = 0.012; Figure 1E). We found, however, no difference between groups in the β-actin normalized levels of pTrkB (% of control 102.3 ± 7.847; *p* = 0.433; Figure 1F), which indicates that the decrease in the pTrkB:mature TrkB ratio was due to an increase in TrkB, and not to reduced phosphorylation of the tyrosine residue.

We found no significant difference in BDNF levels in the HIP of animals eating the CAF (% of control 113.1 ± 24.04; *p* = 0.11; Figure 1G). Similarly, no difference was found in total TrkB levels (% of control 103.5 ± 11.71; *p* = 0.549; Figure 1H), its mature form levels (% of control 94.59 ± 16.99; *p* = 0.319; Figure 1I), its immature form levels (% of control 104.8 ± 11.97; *p* = 0.447; Figure 1J), in the pTrkB to mature TrkB ratio (% of control 94.31 ± 7.119; *p* = 0.649; Figure 1K), or in the pTrkB levels (% of control 94.59 ± 16.99; *p* = 0.319; Figure 1L).

### 3.3. Monoamine and Metabolites

Data are summarized in Table 2. No differences were found between groups in the PFC in the levels of dopamine (*p* = 0.23) and its metabolites DOPAC (*p* = 0.23) and HVA (*p* = 0.09). Similarly, no differences were found between groups in the levels of 5-HT (*p* = 0.07) and its metabolite 5-HIAA (*p* = 0.87) in the PFC and in the HIP (5-HT *p* = 0.7, 5-HIAA *p* = 0.27). No differences were found in the levels of norepinephrine in both PFC (*p* = 0.75) and HIP (*p* = 0.4).

### 3.4. Oxidative Stress Markers

Data are summarized in Table 3. No difference was found in the levels of vitamin C (*p* = 0.602), but the animals fed the CAF displayed both a higher absolute quantity DHA (*p* = 0.014), and a higher ratio of DHA to total vitamin C (*p* = 0.007), indicating an increased turnover of vitamin C, probably due to elevated levels of reactive oxygen species in CAF-fed animals. We also found no difference in uric acid levels between groups (*p* = 0.063). Interestingly, the levels of MDA, a marker of lipid peroxidation [37,40], were lower in animals fed the CAF (*p* = 0.013) compared to the controls. Thus, the suggested redox imbalance did not impose overall peroxidation of cellular lipids significantly. Levels of BH_4_ and its oxidized form BH_2_ were analyzed in the right PFC (Table 4). Animals fed the CAF displayed lower levels of both BH_4_ (*p* = 0.008) and BH_2_ (*p* = 0.013) as well as of the total biopterine concentration (*p* = 0.006). However, the ratio BH_2_:BH_4_ did not differ between groups (*p* = 0.073), thus not suggesting an imbalance of the BH_4_/BH_2_ metabolism.

## 4. Discussion

Collectively, our results suggest an induction of the BDNF-TrkB signaling pathway in the PFC of rats fed a CAF. This is in contrast to previous studies in high-fat-fed rats that were, however, exposed to social isolation and to a traditional high-fat diet [41], which could be the cause of this incongruence. One study involving behavioral trials did not find any effect on the levels of BDNF in the PFC of rats fed a diet rich in fat and sucrose, but found a negative effect in rats fed fat and dextrose, thus suggesting a role of dietary sources on the BDNF metabolism [42]. Another study did not find an effect of a high fat diet on the BDNF mRNA expression in the cortex of high-fat-fed rats [9]. However, no anatomical limits were specified in this work. Additionally, White et al. (2009) did not find any effect on the levels of BDNF protein levels in the cortex of rats born of high-fat-fed dams [43], and a more recent study of Wistar rats fed a CAF did not find any differences compared to control animals [44]. When comparing studies reporting mRNA expression and protein levels, correlation analysis in large-scale datasets reported approximately 50% correspondence between mRNA and protein levels [45,46,47].

The rats in this study were employed in a behavioral study, where they were tested in an operant-conditioning paradigm [32]. Rats fed CAF adopted a more optimal behavioral strategy (more rewards for less work) faster than the controls, which could reflect the elevated levels of cortical BDNF. We did not find differences in levels of BDNF in the HIP. This is in agreement with other studies employing male Sprague-Dawley rats fed a CAF, reporting alterations of BDNF mRNA expression [4,25,27]. Other studies, however, showed reduced levels of hippocampal BDNF in rats fed high fat/high sugar diets, so differences in diet composition and rat strain could account for the discrepancies in the results [9,11].

A limitation of the current study was the absence of a group not exposed to the behavioral paradigm, preventing an assessment of the contributions of the CAF diet and the behavioral challenge on BDNF levels. It can be speculated that the nature of the dietary regimen may have had an “environmental enrichment”-like effect on the rats. The term environmental enrichment generally refers to housing conditions with the addition of toys, cage furniture, and cage mates, which add elements of cognitive, sensory, and motor stimulation [48], and has been reported to have a positive effect on the levels of BDNF in rats [49,50,51]. It has also been reported that a palatable diet reduces stress in rats [28,52]. In turn, stress has been shown to reduce BDNF protein and mRNA levels in the HIP [44,53,54]. However, contrasting results have been reported, probably due to the different experimental conditions used, in terms of source and exposure time of the applied stressor [55,56].

Rats fed CAF showed signs of increased vitamin C turnover in the investigated brain regions. In the brain, vitamin C, in the form of ascorbic acid (AA), functions as a pivotal antioxidant maintaining redox homeostasis by reducing reactive radicals and becoming oxidized to dehydroascorbic acid (DHA) in the process [57,58]. The produced DHA is recycled intracellularly to AA, which can then engage in novel reductions. If the antioxidant (and its recycling) capacity is exceeded, redox imbalance occurs. In turn, this leads to oxidative stress, with negative effects on cellular metabolism, possibly leading to apoptosis and cell death in the brain [59]. At the same time, we found no differences in the levels of uric acid between groups. Uric acid is a natural antioxidant, which has been shown to prevent neuronal apoptosis, and has been used to treat acute ischemic stroke in humans and rats [60,61,62].

Compared to the controls, the rats fed CAF displayed lower levels of MDA, a commonly used marker of lipid peroxidation in the brain [37,40]. It has previously been shown that foods rich in polyphenols have the potential of reducing serum MDA in rats [63]. In addition, coconut oil reduces lipid parameters in the rat and lipid oxidation in vitro as well as MDA in the brains of mice [64,65]. As the foods composing our CAF included high quantities of coconut, this may have contributed to the observed lower MDA levels in the PFC [66]. As we were not able to assess the exact food composition of the applied CAF in the present study due to its multiple components, the food MDA content between groups and its influence on the systemic MDA concentrations cannot be evaluated.

We found no differences in the levels of monoamine neurotransmitters in either PFC or HIP in CAF-fed rats compared to the control counterparts. Both animals and humans experience pleasure from eating palatable foods. For example, rats would endure paw shocks in order to obtain chocolate rewards [67], and humans tend to snack on energy-dense, palatable foods even when not hungry, with a positive correlation with BMI [68]. It has been suggested that obesity and overeating are linked to dysfunctional reward systems, making these conditions comparable to drug addiction (for a review, see [69,70]), supported by data showing that people who are overweight or affected by obesity display increased metabolites of monoamine neurotransmitters in their cerebrospinal fluid [23]. Studies in both mice and rats have shown how a diet rich in fat alters the metabolism of dopamine in the brain (nucleus accumbens) [21], and downregulation of the dopamine D2 receptor [22] and the 5-HT receptors in the HIP of rats [71]. Of these studies, two used different strains [21,22] and only one used a CAF [22], fed to individually-housed rats that were subjected to brain surgery.

The lack of detectable differences observed in our study may be attributed to differences in the experimental setup. Such differences are not an uncommonly encountered issue in preclinical research, and may be a challenge when comparing research findings between experimental setups [72]. Finally, while our results showed lower levels of both BH_4_ and BH_2_ in the PFC of rats fed CAF, no differences in the BH_2_:BH_4_ ratio were found between groups, thus not suggesting an imbalance in the BH_4_/BH_2_ metabolism. BH_4_ has been proposed as a cofactor in the rate-limiting enzyme (tyrosine 3-monooxygenase) in the biosynthesis of dopamine, norepinephrine, and epinephrine [73,74], and the lack of difference in the ratio BH_2_ to BH_4_ agrees with the absence of a difference in neurotransmitter levels. Further studies are required to elucidate how the CAF could affect the metabolism of monoamine neurotransmitters in other brain regions. Moreover, considering the low amount of tissue in the rat HIP, future studies could include subgroups assigned to specific parameters.

In conclusion, these results offer a mixed picture of the effects of CAF on the brain of male Sprague-Dawley rats. Our chronic restricted feeding regimen is linked to positive effects on BDNF levels in the PFC, but the discrepancies with previous reports make a very strong case for moving toward a more rigorous standardization of the concept of the Cafeteria Diet.

## Figures and Tables

**Figure 1 nutrients-14-00126-f001:**
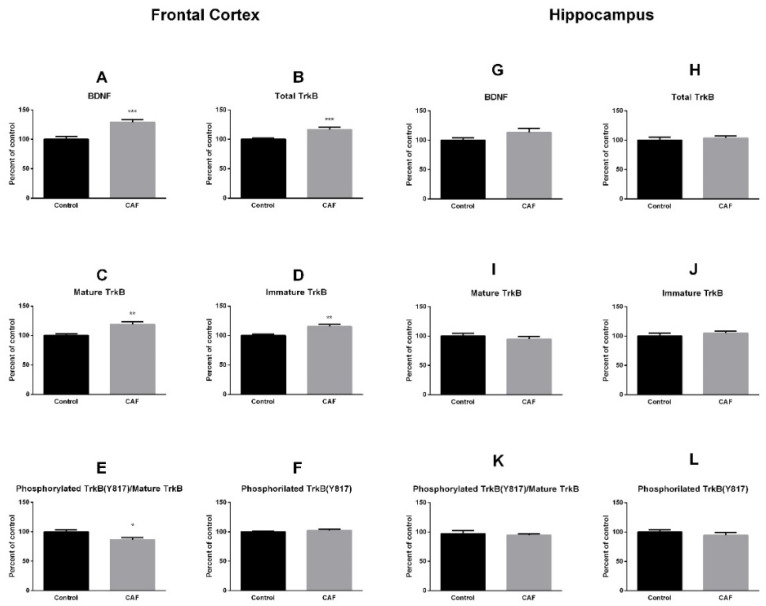
Western blotting results (n = 12 randomly selected samples/diet group) of the levels of BDNF and its receptor TrkB. Data are expressed as percentage relative to the control. All data were normalized to β-actin levels within the same lane/blot, except for the pTrkB:mature TrkB ratio. * *p* < 0.05, ** *p* < 0.01, *** *p* < 0.0005.

**Table 1 nutrients-14-00126-t001:** Nutritional composition of diet treatments.

Per 100 g	Altromin 1324	CAF
Energy (kcal)	318.8	534
Fat (g)	4.1	36.3
Saturated fat (g)	0.46	21.3
Carbohydrates (g)	40.8	47.4
Sugar (g)	4.9	27.4
Protein (g)	19.2	3.8
Water content (g)	10	16

**Table 2 nutrients-14-00126-t002:** Monoamine neurotransmitters and metabolites in the right prefrontal cortex.

	Dopamine		DOPAC		HVA	
Fraction	PFC	HIP	PFC	HIP	PFC	HIP
CAF	7.61 ± 5.29	n/a	5.22 ± 2.92	n/a	1.82 ± 1.97	n/a
Control	9.22 ± 4.76	n/a	6.28 ± 2.70	n/a	0.78 ± 0.49	n/a
	**Serotonin**		**5-HIAA**		**Norepinephrine**	
**Fraction**	**PFC**	**HIP**	**PFC**	**HIP**	**PFC**	**HIP**
CAF	2.20 ± 0.84	0.99 ± 0.19	3.16 ± 0.65	2.88 ± 0.66	1.87 ± 0.31	2.21 ± 0.57
Control	2.65 ± 0.69	1.04 ± 0.30	3.20 ± 0.61	2.64 ± 0.46	1.91 ± 0.43	2.04 ± 0.49

All values are expressed in nmol/g of tissue. “n/a” values refer to the lack of sufficient amounts of hippocampal tissue to obtain readings for the specified parameter. 5-HIAA = 5-hydroxyindoleacetic acid; CAF = Cafeteria Diet; DOPAC = 5,4-dihydroxyphenylacetic acid; HIP = hippocampus; HVA = homovanillic acid; PFC = prefrontal cortex. Data were analyzed by the unpaired t-test and presented as mean ± SD (n = 20/diet group).

**Table 3 nutrients-14-00126-t003:** Markers of oxidative stress in the right prefrontal cortex.

	Vit C	DHA	%DHA of Vit C	Uric Acid	MDA
CAF	2417.00 ± 119.50	96.11 ± 56.97 *	3.99 ± 2.38 **	263.80 ± 21.88	134.30 ± 38.98 *
Control	2486.00 ± 441.30	54.98 ± 42.84	2.16 ± 1.59	253.90 ± 36.76	169.80 ± 46.97

All values are expressed in nmol/g of tissue. DHA = dehydroascorbic acid; MDA = malondialdehyde; CAF = Cafeteria Diet. * *p* < 0.05; ** *p* < 0.01; data were analyzed by the unpaired t-test and presented as mean ± SD (n = 20/diet group).

**Table 4 nutrients-14-00126-t004:** Levels of tetrahydrobiopterine (BH_4_) and its oxidized form dihydrobiopterine (BH_2_) in the right prefrontal cortex.

	BH_4_	BH_2_	Total Biopterin Conc.	BH_2_/BH_4_
CAF	0.444 ± 0.063 **	0.036 ± 0.008 *	0.480 ± 0.067 **	0.081 ± 0.017
Control	0.512 ± 0.089	0.045 ± 0.013	0.557 ± 0.098	0.086 ± 0.021

All values are expressed in nmol/g of tissue, except for the BH2/BH4 ratio. CAF = Cafeteria Diet. * *p* < 0.05; ** *p* < 0.01; data were analyzed by unpaired t-test and presented as mean ± SD (n = 20/diet group).

## Data Availability

All Excel files containing the datasets are available from Figshare. https://doi.org/10.6084/m9.figshare.7532720 (accessed on 20 December 2021); https://doi.org/10.6084/m9.figshare.17695412 (accessed on 20 December 2021); https://doi.org/10.6084/m9.figshare.7532723 (accessed on 20 December 2021).

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
