# Peer review of "A Long-Term Energy-Rich Diet Increases Prefrontal BDNF in Sprague-Dawley Rats"

_nutrients, 2021, doi:10.3390/nu14010126_

Round 1

Reviewer 1 Report

This interesting manuscript deals with the effects of a long-term cafeteria diet on BDNF, monoamine neurotransmitters and oxidative stress markers in the hippocampus and prefrontal cortex of rats.

Surprisingly, Cafeteria diet increased BDNF levels and signaling and reduced lipid peroxidation, highlighting the importance of composition and duration of the dietetic regimen in diet-induced-obesity studies with rodents.

I have only a minor concern about the significant digits of the values given in Tables 2, 3 and 4. Please check and correct.

Author Response

Dear Reviewer 1

Please find our answer to your comment below. Thank you for pointing out this error.

Reviewer 1 wrote: This interesting manuscript deals with the effects of a long-term cafeteria diet on BDNF, monoamine neurotransmitters and oxidative stress markers in the hippocampus and prefrontal cortex of rats. Surprisingly, Cafeteria diet increased BDNF levels and signaling and reduced lipid peroxidation, highlighting the importance of composition and duration of the dietetic regimen in diet-induced-obesity studies with rodents.

 I have only a minor concern about the significant digits of the values given in Tables 2, 3 and 4. Please check and correct.

Answer: We have checked the data and adjusted the number of significant digits/figures used in tables 2, 3 and 4 to two significant digits in table 2 and 3 and three significant digits in table 4 (in which the numbers are all below 1). For norepinephrine/PFC, a typing error was found and corrected (SD is 0.43 in the controls, not 0.31).

Reviewer 2 Report

In the present study, the authors investigated the effects of chronic, restricted, access to CAF on BDNF, monoamine neurotransmitters, and redox imbalance, in HIP and PFC in male rats. They highlighted that CAF could represent a suboptimal feeding regime when investigating the effects of diet-induced obesity in the brain.

The study is well described and the analysis is comprehensive; however, the lack of a group not exposed to the behavioral paradigm represents a limitation.

In the evaluation of the manuscript the following technical point needs to be improved:

1) What is the exact composition of the Cafeteria Diet fed to the animals? The diet has to be described in detail in a separate table or the supplement. 

Author Response

Dear Reviewer 2

Please find our answer to your comment below. Thank you for pointing out this lack of necessary information

 Reviewer 2 wrote: In the present study, the authors investigated the effects of chronic, restricted, access to CAF on BDNF, monoamine neurotransmitters, and redox imbalance, in HIP and PFC in male rats. They highlighted that CAF could represent a suboptimal feeding regime when investigating the effects of diet-induced obesity in the brain. The study is well described and the analysis is comprehensive; however, the lack of a group not exposed to the behavioral paradigm represents a limitation.

 In the evaluation of the manuscript the following technical point needs to be improved:

1) What is the exact composition of the Cafeteria Diet fed to the animals? The diet has to be described in detail in a separate table or the supplement.

Answer: We have added a supplementary table with the four constituents of the CAF diet added to the main table in the manuscript; Table 1. Moreover, we have revised the number of significant digits in main table 1 following up on the comments on significant digits from Reviewer 1.